# Learning Energy-Based Models in High-Dimensional Spaces with Multi-scale Denoising Score Matching

## Abstract

Energy-Based Models (EBMs) assign unnormalized log-probability to data samples. This functionality has a variety of applications, such as sample synthesis, data denoising, sample restoration, outlier detection, Bayesian reasoning, and many more. But training of EBMs using standard maximum likelihood is extremely slow because it requires sampling from the model distribution. Score matching potentially alleviates this problem. In particular, denoising score matching (Vincent, 2011) has been successfully used to train EBMs. Using noisy data samples with one fixed noise level, these models learn fast and yield good results in data denoising (Saremi and Hyvarinen, 2019). However, demonstrations of such models in high quality sample synthesis of high dimensional data were lacking. Recently, Song and Ermon (2019) have shown that a generative model trained by denoising score matching accomplishes excellent sample synthesis, when trained with data samples corrupted with multiple levels of noise. Here we provide analysis and empirical evidence showing that training with multiple noise levels is necessary when the data dimension is high. Leveraging this insight, we propose a novel EBM trained with multi-scale denoising score matching. Our model exhibits data generation performance comparable to state-of-the-art techniques such as (Song and Ermon, 2019) and GANs, and sets a new baseline for EBMs. The proposed model also provides density information and performs well in an image inpainting task.

## 1 Introduction and Motivation

Treating data as stochastic samples from a probability distribution and developing models that can learn such distributions is at the core for solving a large variety of application problems, such as error correction/denoising (Vincent et al., 2010), outlier/novelty detection (Zhai et al., 2016; Choi and Jang, 2018), sample generation (Nijkamp et al., 2019; Du and Mordatch, 2019), invariant pattern recognition, Bayesian reasoning (Welling and Teh, 2011) which relies on good data priors, and many others.

Energy-Based Models (EBMs) (LeCun et al., 2006; Ngiam et al., 2011) assign an energy $E(\boldsymbol{x})$ to each data point $\boldsymbol{x}$ which implicitly defines a probability by the Boltzmann distribution $p_m(\boldsymbol{x}) = e^{-E(\boldsymbol{x})}/Z$. Sampling from this distribution can be used as a generative process that yield plausible samples of $\boldsymbol{x}$. Compared to other generative models, like GANs (Goodfellow et al., 2014), flow-based models (Dinh et al., 2015; Kingma and Dhariwal, 2018), or auto-regressive models (van den Oord et al., 2016; Ostrovski et al., 2018), energy-based models have significant advantages. First, they provide explicit (unnormalized) density information, compositionality (Hinton, 1999; Haarnoja et al., 2017), better mode coverage (Kumar et al., 2019) and flexibility (Du and Mordatch, 2019). Further, they do not require special model architecture, unlike auto-regressive and flow-based models. Recently, Energy-based models has been successfully trained with maximum likelihood (Nijkamp et al., 2019; Du and Mordatch, 2019), but training can be very computationally demanding due to the need of sampling model distribution. Variants with a truncated sampling procedure have been proposed, such as contrastive divergence (Hinton, 2002). Such models learn much faster with the draw back of not exploring the state space thoroughly (Tieleman, 2008).

## 1.1 Score Matching, Denoising Score Matching and Deep Energy Estimators

*Score matching* (SM) (Hyvärinen, 2005) circumvents the requirement of sampling the model distribution. In score matching, the score function is defined to be the gradient of log-density or the negative energy function. The expected $L2$ norm of difference between the model score function and the data score function are minimized. One convenient way of using score matching is learning the energy function corresponding to a Gaussian kernel Parzen density estimator (Parzen, 1962) of the data: $p_{\sigma_0}(\tilde{\boldsymbol{x}}) = \int q_{\sigma_0}(\tilde{\boldsymbol{x}}|\boldsymbol{x})p(\boldsymbol{x})d\boldsymbol{x}$. Though hard to evaluate, the data score is well defined: $s_d(\tilde{\boldsymbol{x}}) = \nabla_{\tilde{\boldsymbol{x}}} \log(p_{\sigma_0}(\tilde{\boldsymbol{x}}))$, and the corresponding objective is:

$$L_{SM}(\theta) = \mathbb{E}_{p_{\sigma 0}(\tilde{\boldsymbol{x}})} \parallel \nabla_{\tilde{\boldsymbol{x}}} \log(p_{\sigma_0}(\tilde{\boldsymbol{x}})) + \nabla_{\tilde{\boldsymbol{x}}} E(\tilde{\boldsymbol{x}}; \theta) \parallel^2 \tag{1}$$

Vincent (2011) studied the connection between denoising auto-encoder and score matching, and proved the remarkable result that the following objective, named *Denoising Score Matching* (DSM), is equivalent to the objective above:

$$L_{DSM}(\theta) = \mathbb{E}_{p_{\sigma_0}(\tilde{\boldsymbol{x}},\boldsymbol{x})} \parallel \nabla_{\tilde{\boldsymbol{x}}} \log(q_{\sigma 0}(\tilde{\boldsymbol{x}}|\boldsymbol{x})) + \nabla_{\tilde{\boldsymbol{x}}} E(\tilde{\boldsymbol{x}}; \theta) \parallel^2 \tag{2}$$

Note that in (2) the Parzen density score is replaced by the derivative of log density of the single noise kernel $\nabla_{\tilde{\boldsymbol{x}}} \log(q_{\sigma_0}(\tilde{\boldsymbol{x}}|\boldsymbol{x}))$, which is much easier to evaluate. In the particular case of Gaussian noise, $\log(q_{\sigma_0}(\tilde{\boldsymbol{x}}|\boldsymbol{x})) = -\frac{(\tilde{\boldsymbol{x}}-\boldsymbol{x})^2}{2\sigma_0^2} + C$, and therefore:

$$L_{DSM}(\theta) = \mathbb{E}_{p_{\sigma 0}(\tilde{\boldsymbol{x}},\boldsymbol{x})} \parallel \boldsymbol{x} - \tilde{\boldsymbol{x}} + {\sigma_0}^2 \nabla_{\tilde{\boldsymbol{x}}} E(\tilde{\boldsymbol{x}}; \theta) \parallel^2 \tag{3}$$

The interpretation of objective (3) is simple, it forces the energy gradient to align with the vector pointing from the noisy sample to the clean data sample. To optimize an objective involving the derivative of a function defined by a neural network, Kingma and LeCun (2010) proposed the use of double backpropagation (Drucker and Le Cun, 1991). *Deep energy estimator networks* (Saremi et al., 2018) first applied this technique to learn an energy function defined by a deep neural network. In this work and similarly in Saremi and Hyvarinen (2019), an energy-based model was trained to match a Parzen density estimator of data with a certain noise magnitude. The previous models were able to perform denoising task, but they were unable to generate high-quality data samples from a random input initialization. Recently, Song and Ermon (2019) trained an excellent generative model by fitting a series of score estimators coupled together in a single neural network, each matching the score of a Parzen estimator with a different noise magnitude.

The questions we address here is why learning energy-based models with single noise level does not permit high-quality sample generation and what can be done to improve energy based models. Our work builds on key ideas from Saremi et al. (2018); Saremi and Hyvarinen (2019); Song and Ermon (2019). Section 2, provides a geometric view of the learning problem in denoising score matching and provides a theoretical explanation why training with one noise level is insufficient if the data dimension is high. Section 3 presents a novel method for training energy based model, *Multiscale Denoising Score Matching* (MDSM). Section 4 describes empirical results of the MDSM model and comparisons with other models.

## 2 A Geometric view of Denoising Score Matching

Song and Ermon (2019) used denoising score matching with a range of noise levels, achieving great empirical results. The authors explained that large noise perturbation are required to enable the learning of the score in low-data density regions. But it is still unclear why a series of different noise levels are necessary, rather than one single large noise level. Following Saremi and Hyvarinen (2019), we analyze the learning process in denoising score matching based on measure concentration properties of high-dimensional random vectors.

We adopt the common assumption that the data distribution to be learned is high-dimensional, but only has support around a relatively low-dimensional manifold (Tenenbaum et al., 2000; Roweis and Saul, 2000; Lawrence, 2005). If the assumption holds, it causes a problem for score matching: The density, or the gradient of the density is then undefined outside the manifold, making it difficult

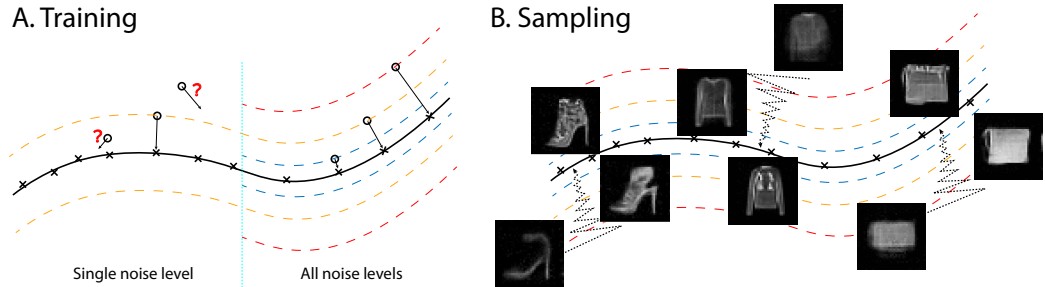

Figure 1: Illustration of anneal denoising score matching. A. During training, derivative of log-likelihood is forced to point toward data manifold, establishing energy difference between points within manifold and points outside. Note that energy is negative log-likelihood therefore energy is higher for point further away from data manifold. B. During annealed Langevin sampling, sample travel from outside data manifold to data manifold. Shown are singled step denoised sample during sampling of an energy function trained with MDSM on Fashion-MNIST (see text for details).

to train a valid density model for the data distribution defined on the entire space. Saremi and Hyvarinen (2019) and Song and Ermon (2019) discussed this problem and proposed to smooth the data distribution with a Gaussian kernel to alleviate the issue.

To further understand the learning in denoising score matching when the data lie on a manifold $\mathcal{X}$ and the data dimension is high, two elementary properties of random Gaussian vectors in high-dimensional spaces are helpful: First, the length distribution of random vectors becomes concentrated at $\sqrt{d}\sigma$ (Vershynin, 2018), where $\sigma^2$ is the variance of a single dimension. Second, a random vector is always close to orthogonal to a fixed vector (Tao, 2012). With these premises one can visualize the configuration of noisy and noiseless data points that enter the learning process: A data point $\boldsymbol{x}$ sampled from $\mathcal{X}$ and its noisy version $\tilde{\boldsymbol{x}}$ always lie on a line which is almost perpendicular to the tangent space $T_{\boldsymbol{x}}\mathcal{X}$ and intersects $\mathcal{X}$ at $\boldsymbol{x}$. Further, the distance vectors between $(\boldsymbol{x}, \tilde{\boldsymbol{x}})$ pairs all have similar length $\sqrt{d}\sigma$. As a consequence, the set of noisy data points concentrate on a set $\tilde{\mathcal{X}}_{\sqrt{d}\sigma,\epsilon}$ that has a distance with $(\sqrt{d}\sigma - \epsilon, \sqrt{d}\sigma + \epsilon)$ from the data manifold $\mathcal{X}$, where $\epsilon \ll \sqrt{d}\sigma$.

Therefore, performing denoising score matching learning with $(\boldsymbol{x}, \tilde{\boldsymbol{x}})$ pairs generated with a fixed noise level $\sigma$, which is the approach taken previously except in Song and Ermon (2019), will match the score in the set $\tilde{\mathcal{X}}_{\sqrt{d}\sigma,\epsilon}$ and enable denoising of noisy points in the same set. However, the learning provides little information about the density outside this set, farther or closer to the data manifold, as noisy samples outside $\tilde{\mathcal{X}}_{\sqrt{d}\sigma,\epsilon}$ rarely appear in the training process. An illustration is presented in Figure 1A.

Let $\tilde{\mathcal{X}}^C_{\sqrt{d}\sigma,\epsilon}$ denote the complement of the set $\tilde{\mathcal{X}}_{\sqrt{d}\sigma,\epsilon}$. Even if $p_{\sigma_0}(\tilde{\boldsymbol{x}} \in \tilde{\mathcal{X}}^C_{\sqrt{d}\sigma,\epsilon})$ is very small in high-dimensional space, the score in $\tilde{\mathcal{X}}^C_{\sqrt{d}\sigma,\epsilon}$ still plays a critical role in sampling from random initialization. This analysis may explain why models based on denoising score matching, trained with a single noise level encounter difficulties in generating data samples when initialized at random. For an empirical support of this explanation, see our experiments with models trained with single noise magnitudes (Appendix B). To remedy this problem, one has to apply a learning procedure of the sort proposed in Song and Ermon (2019), in which samples with different noise levels are used. Depending on the dimension of the data, the different noise levels have to be spaced narrowly enough to avoid empty regions in the data space. In the following, we will use Gaussian noise and employ a Gaussian scale mixture to produce the noisy data samples for the training (for details, See Section 3.1 and Appendix A).

Another interesting property of denoising score matching was suggested in the denoising auto-encoder literature (Vincent et al., 2010; Karklin and Simoncelli, 2011). With increasing noise level, the learned features tend to have larger spatial scale. In our experiment we observe similar phenomenon when training model with denoising score matching with single noise scale. If one compare samples in Figure B.1, Appendix B, it is evident that noise level of 0.3 produced a model that learned short range correlation that spans only a few pixels, noise level of 0.6 learns longer stroke

structure without coherent overall structure, and noise level of 1 learns more coherent long range structure without details such as stroke width variations. This suggests that training with single noise level in denoising score matching is not sufficient for learning a model capable of high-quality sample synthesis, as such a model have to capture data structure of all scales.

# 3 LEARNING ENERGY-BASED MODEL WITH MULTISCALE DENOISING SCORE MATCHING

## 3.1 MULTISCALE DENOISING SCORE MATCHING

Motivated by the analysis in section 2, we strive to develop an EBM based on denoising score matching that can be trained with noisy samples in which the noise level is not fixed but drawn from a distribution. The model should approximate the Parzen density estimator of the data $p_{\sigma_0}(\tilde{\boldsymbol{x}}) = \int q_{\sigma_0}(\tilde{\boldsymbol{x}}|\boldsymbol{x})p(\boldsymbol{x})dx$. Specifically, the learning should minimize the difference between the derivative of the energy and the score of $p_{\sigma_0}$ under the expectation $\mathbb{E}_{p_M(\tilde{\boldsymbol{x}})}$ rather than $\mathbb{E}_{p_{\sigma_0}(\tilde{\boldsymbol{x}})}$, the expectation taken in standard denoising score matching. Here $p_M(\tilde{\boldsymbol{x}}) = \int q_M(\tilde{\boldsymbol{x}}|\boldsymbol{x})p(\boldsymbol{x})dx$ is chosen to cover the signal space more evenly to avoid the measure concentration issue described above. The resulting *Multiscale Score Matching* (MSM) objective is:

$$L_{MSM}(\theta) = \mathbb{E}_{p_M(\tilde{\boldsymbol{x}})} \parallel \nabla_{\tilde{\boldsymbol{x}}} \log(p_{\sigma_0}(\tilde{\boldsymbol{x}})) + \nabla_{\tilde{\boldsymbol{x}}} E(\tilde{\boldsymbol{x}}; \theta) \parallel^2 \tag{4}$$

Compared to the objective of denoising score matching (1), the only change in the new objective (4) is the expectation. Both objectives are consistent, if $p_M(\tilde{\boldsymbol{x}})$ and $p_{\sigma_0}(\tilde{\boldsymbol{x}})$ have the same support, as shown formally in Proposition 1 of Appendix A. In Proposition 2, we prove that Equation 4 is equivalent to the following denoising score matching objective:

$$L_{MDSM^*} = \mathbb{E}_{p_M(\tilde{\boldsymbol{x}})q_{\sigma_0}(\boldsymbol{x}|\tilde{\boldsymbol{x}})} \parallel \nabla_{\tilde{\boldsymbol{x}}} \log(q_{\sigma 0}(\tilde{\boldsymbol{x}}|\boldsymbol{x})) + \nabla_{\tilde{\boldsymbol{x}}} E(\tilde{\boldsymbol{x}}; \theta) \parallel^2 \tag{5}$$

The above results hold for any noise kernel $q_{\sigma_0}(\tilde{\boldsymbol{x}}|\boldsymbol{x})$, but Equation 5 contains the reversed expectation, which is difficult to evaluate in general. To proceed, we choose $q_{\sigma_0}(\tilde{\boldsymbol{x}}|\boldsymbol{x})$ to be Gaussian, and also choose $q_M(\tilde{\boldsymbol{x}}|\boldsymbol{x})$ to be a Gaussian scale mixture: $q_M(\tilde{\boldsymbol{x}}|\boldsymbol{x}) = \int q_{\sigma}(\tilde{\boldsymbol{x}}|\boldsymbol{x})p(\sigma)d\sigma$ and $q_{\sigma}(\tilde{\boldsymbol{x}}|\boldsymbol{x}) = \mathcal{N}(\boldsymbol{x}, \sigma^2 I_d)$. After algebraic manipulation and one approximation (see the derivation following Proposition 2 in Appendix A), we can transform Equation 5 into a more convenient form, which we call *Multiscale Denoising Score Matching* (MDSM):

$$L_{MDSM} = \mathbb{E}_{p(\sigma)q_{\sigma}(\tilde{\boldsymbol{x}}|\boldsymbol{x})p(\boldsymbol{x})} \parallel \nabla_{\tilde{\boldsymbol{x}}} \log(q_{\sigma 0}(\tilde{\boldsymbol{x}}|\boldsymbol{x})) + \nabla_{\tilde{\boldsymbol{x}}} E(\tilde{\boldsymbol{x}}; \theta) \parallel^2 \tag{6}$$

The square loss term evaluated at noisy points $\tilde{\boldsymbol{x}}$ at larger distances from the true data points $\boldsymbol{x}$ will have larger magnitude. Therefore, in practice it is convenient to add a monotonically decreasing term $l(\sigma)$ for balancing the different noise scales, e.g. $l(\sigma) = \frac{1}{\sigma^2}$. Ideally, we want our model to learn the correct gradient everywhere, so we would need to add noise of all levels. However, learning denoising score matching at very large or very small noise levels is useless. At very large noise levels the information of the original sample is completely lost. Conversely, in the limit of small noise, the noisy sample is virtually indistinguishable from real data. In neither case one can learn a gradient which is informative about the data structure. Thus, the noise range needs only to be broad enough to encourage learning of data features over all scales. Particularly, we do not sample $\sigma$ but instead choose a series of fixed $\sigma$ values $\sigma_1 \cdots \sigma_K$. Further, substituting $\log(q_{\sigma_0}(\tilde{\boldsymbol{x}}|\boldsymbol{x})) = -\frac{(\tilde{\boldsymbol{x}}-\boldsymbol{x})^2}{2\sigma_0^2} + C$ into Equation 4, we arrive at the final objective:

$$L(\theta) = \sum_{\sigma \in \{\sigma_1 \cdots \sigma_K\}} \mathbb{E}_{q_{\sigma}(\tilde{\boldsymbol{x}}|\boldsymbol{x})p(\boldsymbol{x})} l(\sigma) \parallel \boldsymbol{x} - \tilde{\boldsymbol{x}} + \sigma_0^2 \nabla_{\tilde{\boldsymbol{x}}} E(\tilde{\boldsymbol{x}}; \theta) \parallel^2 \tag{7}$$

It may seem that $\sigma_0$ is an important hyperparameter to our model, but after our approximation $\sigma_0$ become just a scaling factor in front of the energy function, and can be simply set to one as long as the temperature range during sampling is scaled accordingly (See Section 3.2). Therefore the only hyper-parameter is the rang of noise levels used during training.

On the surface, objective (7) looks similar to the one in Song and Ermon (2019). The important difference is that Equation 7 approximates a *single* distribution, namely $p_{\sigma_0}(\tilde{\boldsymbol{x}})$, the data smoothed with one fixed kernel $q_{\sigma_0}(\tilde{\boldsymbol{x}}|\boldsymbol{x})$. In contrast, Song and Ermon (2019) approximate the score of *multiple* distributions, the family of distributions $\{p_{\sigma_i}(\tilde{\boldsymbol{x}}) : i = 1, ..., n\}$, resulting from the data smoothed by kernels of different widths $\sigma_i$. Because our model learns only a single target distribution, it does not require noise magnitude as input.

## 3.2 Sampling by Annealed Langevin Dynamics

Langevin dynamics has been used to sample from neural network energy functions (Du and Mordatch, 2019; Nijkamp et al., 2019). However, the studies described difficulties with mode exploration unless very large number of sampling steps is used. To improve mode exploration, we propose incorporating simulated annealing in the Langevin dynamics. Simulated annealing (Kirkpatrick et al., 1983; Neal, 2001) improves mode exploration by sampling first at high temperature and then cooling down gradually. This has been successfully applied to challenging computational problems, such as combinatorial optimization.

To apply simulated annealing to Langevin dynamics. Note that in a model of Brownian motion of a physical particle, the temperature in the Langevin equation enters as a factor $\sqrt{T}$ in front of the noise term, some literature uses $\sqrt{\beta^{-1}}$ where $\beta = 1/T$ (Jordan et al., 1998). Adopting the $\sqrt{T}$ convention, the Langevin sampling process (Bellec et al., 2017) is given by:

$$\boldsymbol{x}_{t+1} = \boldsymbol{x}_t - \frac{\epsilon^2}{2}\nabla_{\boldsymbol{x}}E(\boldsymbol{x}_t; \theta) + \epsilon\sqrt{T_t}\mathcal{N}(0, I_d) \tag{8}$$

where $T_t$ follows some annealing schedule, and $\epsilon$ denotes step length, which is fixed. During sampling, samples behave very much like physical particles under Brownian motion in a potential field. Because the particles have average energies close to the their current thermic energy, they explore the state space at different distances from data manifold depending on temperature. Eventually, they settle somewhere on the data manifold. The behavior of the particle's energy value during a typical annealing process is depicted in Appendix Figure F.1B.

If the obtained sample is still slightly noisy, we can apply a single step gradient denoising jump (Saremi and Hyvarinen, 2019) to improve sample quality:

$$\boldsymbol{x}_{clean} = \boldsymbol{x}_{noisy} - \sigma_0^2\nabla_{\boldsymbol{x}}E(\boldsymbol{x}_{noisy}; \theta) \tag{9}$$

This denoising procedure can be applied to noisy sample with any level of Gaussian noise because in our model the gradient automatically has the right magnitude to denoise the sample. This process is justified by the Empirical Bayes interpretation of this denoising process, as studied in Saremi and Hyvarinen (2019).

Song and Ermon (2019) also call their sample generation process annealed Langevin dynamics. It should be noted that their sampling process does not coincide with Equation 8. Their sampling procedure is best understood as sequentially sampling a series of distributions corresponding to data distribution corrupted by different levels of noise.

## 4 Image Modeling Results

**Training and Sampling Details.** The proposed energy-based model is trained on standard image datasets, specifically MNIST, Fashion MNIST, CelebA (Liu et al., 2015) and CIFAR-10 (Krizhevsky et al., 2009). During training we set $\sigma_0 = 0.1$ and train over a noise range of $\sigma \in [0.05, 1.2]$, with the different noise uniformly spaced on the batch dimension. For MNIST and Fashion MNIST we used geometrically distributed noise in the range $[0.1, 3]$. The weighting factor $l(\sigma)$ is always set to $1/\sigma^2$ to make the square term roughly independent of $\sigma$. We fix the batch size at 128 and use the Adam optimizer with a learning rate of $5 \times 10^{-5}$. For MNIST and Fashion MNIST, we use a 12-Layer ResNet with 64 filters, for the CelebA and CIFAT-10 data sets we used a 18-Layer ResNet with 128 filters (He et al., 2016a;b). No normalization layer was used in any of the networks. We designed the output layer of all networks to take a generalized quadratic form (Fan et al., 2018). Because the energy function is anticipated to be approximately quadratic with respect to the noise level, this modification was able to boost the performance significantly. For more detail on training

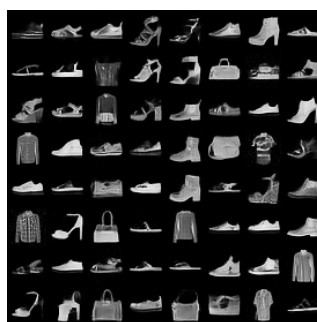 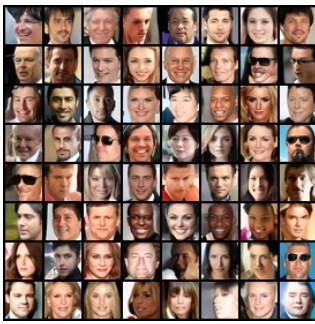 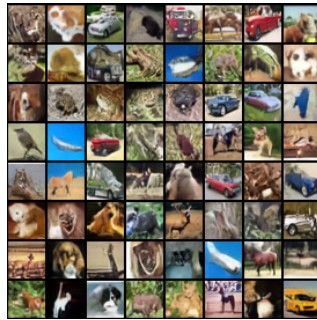

Figure 2: Samples from our model trained on Fashion MNIST, CelebA and CIFAR-10. See Figure E.3 and Figure E.4 in Appendix for more samples and comparison with training data.

and model architecture, see Appendix D. One notable result is that since our training method does not involve sampling, we achieved a speed up of roughly an order of magnitude compared to the maximum-likelihood training using Langevin dynamics [1]. Our method thus enables the training of energy-based models even when limited computational resources prohibit maximum likelihood methods.

We found that the choice of the maximum noise level has little effect on learning as long as it is large enough to encourage learning of the longest range features in the data. However, as expected, learning with too small or too large noise levels is not beneficial and can even destabilize the training process. Further, our method appeared to be relatively insensitive to how the noise levels are distributed over a chosen range. Geometrically spaced noise as in (Song and Ermon, 2019) and linearly spaced noise both work, although in our case learning with linearly spaced noise was somewhat more robust.

For sampling the learned energy function we used annealed Langevin dynamics with an empirically optimized annealing schedule, see Figure F.1B for the particular shape of annealing schedule we used. In contrast, annealing schedules with theoretical guaranteed convergence property takes extremely long (Geman and Geman, 1984). The range of temperatures to use in the sampling process depends on the choice of $\sigma_0$, as the equilibrium distribution is roughly images with Gaussian noise of magnitude $\sqrt{T}\sigma_0$ added on top. To ease traveling between modes far apart and ensure even sampling, the initial temperature needs to be high enough to inject noise of sufficient magnitude. A choice of $T = 100$, which corresponds to added noise of magnitude $\sqrt{100} * 0.1 = 1$, seems to be sufficient starting point. For step length $\epsilon$ we generally used $0.02$, although any value within the range $[0.015, 0.05]$ seemed to work fine. After the annealing process we performed a single step denoising to further enhance sample quality.

Table 1: Unconditional Inception score, FID scores and Likelihoods for CIFAR-10

| Model | IS | FID | Likelihood | NNL (bits/dim) |
|---|---|---|---|---|
| iResNet (Behrmann et al., 2019) | - | 65.01 | Yes | 3.45 |
| PixelCNN (van den Oord et al., 2016) | 4.60 | 65.93 | Yes | **3.14** |
| PixelIQN (Ostrovski et al., 2018) | 5.29 | 49.46 | Yes | - |
| Residual Flow (Chen et al., 2019) | - | 46.37 | Yes | 3.28 |
| GLOW (Kingma and Dhariwal, 2018) | - | 46.90 | Yes | 3.35 |
| EBM (ensemble) (Du and Mordatch, 2019) | 6.78 | 38.2 | Yes(density) | - [2] |
| SNGAN (Miyato et al., 2018) | 8.22 | **21.7** | No | - |
| MDSM (Ours) | 8.31 | 31.7 | Yes(density) | -0.96 [3] |
| NCSN (Song and Ermon, 2019) | **8.91** | 25.32 | No | - |

---

[1] For example, on a single GPU, training MNIST with a 12-layer Resnet takes 0.3s per batch with our method, while maximum likelihood training with a modest 30 Langevin step per weight update takes 3s per batch. Both methods need similar number of weight updates to train.

**Unconditional Image Generation.** We demonstrate the generative ability of our model by displaying samples obtained by annealed Langevin sampling and single step denoising jump. We evaluated 50k sampled images after training on CIFAR-10 with two performance scores, Inception (Salimans et al., 2016) and FID (Heusel et al., 2017). We achieved Inception Score of 8.31 and FID of 31.7, comparable to modern GAN approaches. Scores for CelebA dataset are not reported here as they are not commonly reported and may depend on the specific pre-processing used. More samples and training images are provided in Appendix for visual inspection. We believe that visual assessment is still essential because of the possible issues with the Inception score (Barratt and Sharma, 2018). Indeed, we also found that the visually impressive samples were not necessarily the one achieving the highest Inception Score.

Although overfitting is not a common concern for generative models, we still tested our model for overfitting. We found no indication for overfitting by comparing model samples with their nearest neighbors in the data set, see Figure C.1 in Appendix.

**Mode Coverage.** We repeated with our model the 3 channel MNIST mode coverage experiment similar to the one in Kumar et al. (2019). An energy-based model was trained on 3-channel data where each channel is a random MNIST digit. Then 8000 samples were taken from the model and each channel was classified using a small MNIST classifier network. We obtained results of the 966 modes, comparable to GAN approaches. Training was successful and our model assigned low energy to all the learned modes, but some modes were not accessed during sampling, likely due to the Langevin Dynamics failing to explore these modes. A better sampling technique such as HMC Neal et al. (2011) or a Maximum Entropy Generator (Kumar et al., 2019) could improve this result.

**Image Inpainting.** Image impainting can be achieved with our model by clamping a part of the image to ground truth and performing the same annealed Langevin and Jump sampling procedure on the missing part of the image. Noise appropriate to the sampling temperature need to be added to the clamped inputs. The quality of inpainting results of our model trained on CelebA and CIFAR-10 can be assessed in Figure 3. For CIFAR-10 inpainting results we used the test set.

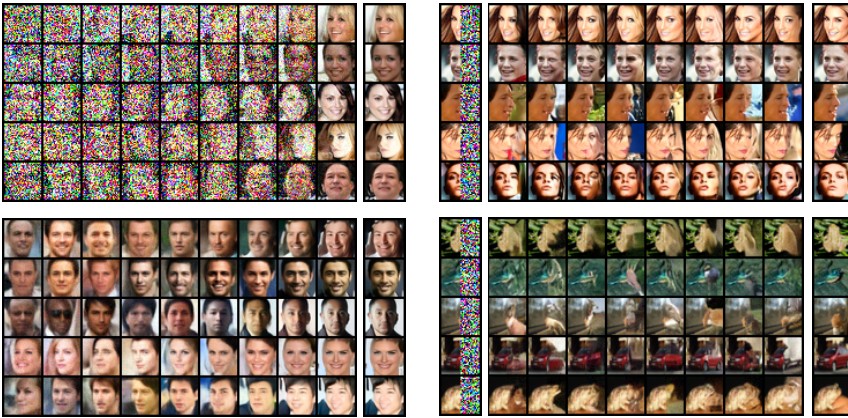

Figure 3: Demonstration of the sampling process (left), and image inpainting (right). The sampling process is shown with Gaussian noise (top left), and denoised by single step gradient jump (lower left). The column next to sampling process shows samples after the last denoising jump at the end of sampling. Inpainting results are shown next to initial image (left column) and the ground truth image (right column).

**Log likelihood estimation.** For energy-based models, the log density can be obtained after estimating the partition function with Annealed Importance Sampling (AIS) (Salakhutdinov and Murray, 2008) or Reverse AIS (Burda et al., 2015). In our experiment on CIFAR-10 model, similar to reports in Du and Mordatch (2019), there is still a substantial gap between AIS and Reverse AIS estimation, even after very substantial computational effort. In Table 1, we report result from Reverse AIS, as it tends to over-estimate the partition function thus underestimate the density. Note that although

---

[2]Author reported difficulties evaluating Likelihood
[3]Upper Bound obtained by Reverse AIS

density values and likelihood values are not directly comparable, we list them together due to the sheer lack of a density model for CIFAR-10.

We also report a density of 1.21 bits/dim on MNIST dataset, and we refer readers to Du and Mordatch (2019) for comparison to other models on this dataset. More details on this experiment is provided in the Appendix.

**Outlier Detection.** Choi and Jang (2018) and Nalisnick et al. (2019) have reported intriguing behavior of high dimensional density models on out of distribution samples. Specifically, they showed that a lot of models assign higher likelihood to out of distribution samples than real data samples. We investigated whether our model behaves similarly.

Our energy function is only trained outside the data manifold where samples are noisy, so the energy value at clean data points may not always be well behaved. Therefore, we added noise with magnitude $\sigma_0$ before measuring the energy value. We find that our network behaves similarly to previous likelihood models, it assigns lower energy, thus higher density, to some OOD samples. We show one example of this phenomenon in Appendix Figure F.1A.

We also attempted to use the denoising performance, or the objective function to perform outlier detection. Intriguingly, the results are similar as using the energy value. Denoising performance seems to correlate more with the variance of the original image than the content of the image.

## 5 DISCUSSION

In this work we provided analyses and empirical results for understanding the limitations of learning the structure of high-dimensional data with denoising score matching. We found that the objective function confines learning to a small set due to the measure concentration phenomenon in random vectors. Therefore, sampling the learned distribution outside the set where the gradient is learned does not produce good result. One remedy to learn meaningful gradients in the entire space is to use samples during learning that are corrupted by different amounts of noise. Indeed, Song and Ermon (2019) applied this strategy very successfully.

The central contribution of our paper is to investigate how to use a similar learning strategy in EBMs. Specifically, we proposed a novel EBM model, the *Multiscale Denoising Score Matching* (MDSM) model. The new model is capable of denoising, producing high-quality samples from random noise, and performing image inpainting. While also providing density information, our model learns an order of magnitude faster than models based on maximum likelihood.

Our approach is conceptually similar to the idea of combining denoising autoencoder and annealing (Geras and Sutton, 2015; Chandra and Sharma, 2014; Zhang and Zhang, 2018) though this idea was proposed in the context of pre-training neural networks for classification applications. Previous efforts of learning energy-based models with score matching (Kingma and LeCun, 2010; Song et al., 2019) were either computationally intensive or unable to produce high-quality samples comparable to those obtained by other generative models such as GANs. Saremi et al. (2018) and Saremi and Hyvarinen (2019) trained energy-based model with the denoising score matching objective but the resulting models cannot perform sample synthesis from random noise initialization.

Recently, Song and Ermon (2019) proposed the NCSN model, capable of high-quality sample synthesis. This model approximates the score of a family of distributions obtained by smoothing the data by kernels of different widths. The sampling in the NCSN model starts with sampling the distribution obtained with the coarsest kernel and successively switches to distributions obtained with finer kernels. Unlike NCSN, our method learns an energy-based model corresponding to $p_{\sigma_0}(\tilde{\boldsymbol{x}})$ for a fixed $\sigma_0$. This method improves score matching in high-dimensional space by matching the gradient of an energy function to the score of $p_{\sigma_0}(\tilde{\boldsymbol{x}})$ in a set that avoids measure concentration issue.

All told, we offer a novel EBM model that achieves high-quality sample synthesis, which among other EBM approaches provides a new state-of-the art. Compared to the NCSN model, our model is more parsimonious than NCSN and can support single step denoising without prior knowledge of the noise magnitude. But our model performs sightly worse than the NCSN model, which could have several reasons. First, the derivation of Equation 6 requires an approximation to keep the training procedure tractable, which could reduce the performance. Second, the NCSNs output is a vector that, at least during optimization, does not always have to be the derivative of a scalar function. In

contrast, in our model the network output is a scalar function. Thus it is possible that the NCSN model performs better because it explores a larger set of functions during optimization.

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

## A  MDSM OBJECTIVE

In this section, we provide a formal discussion of the MDSM objective and suggest it as an improved score matching formulation in high-dimensional space.

Vincent (2011) illustrated the connection between the model score $-\nabla_{\tilde{x}} E(\tilde{x}; \theta)$ with the score of Parzen window density estimator $\nabla_{\tilde{x}} \log(p_{\sigma_0}(\tilde{x}))$. Specifically, the objective is Equation 1 which we restate here:

$$L_{SM}(\theta) = \mathbb{E}_{p_{\sigma 0}(\tilde{x})} \parallel \nabla_{\tilde{x}} \log(p_{\sigma_0}(\tilde{x})) + \nabla_{\tilde{x}} E(\tilde{x}; \theta) \parallel^2 \tag{10}$$

Our key observation is: in high-dimensional space, due the concentration of measure, the expectation w.r.t. $p_{\sigma 0}(\tilde{x})$ over weighs a thin shell at roughly distance $\sqrt{d}\sigma$ to the empirical distribution $p(x)$. Though in theory this is not a problem, in practice this leads to results that the score are only well matched on this shell. Based on this observation, we suggest to replace the expectation w.r.t. $p_{\sigma 0}(\tilde{x})$ with a distribution $p_{\sigma'}(\tilde{x})$ that has the same support as $p_{\sigma 0}(\tilde{x})$ but can avoid the measure concentration problem. We call this *multiscale score matching* and the objective is the following:

$$L_{MSM}(\theta) = \mathbb{E}_{p_M(\tilde{x})} \parallel \nabla_{\tilde{x}} \log(p_{\sigma_0}(\tilde{x})) + \nabla_{\tilde{x}} E(\tilde{x}; \theta) \parallel^2 \tag{11}$$

**Proposition 1.** $L_{MSM}(\theta) = 0 \iff L_{SM}(\theta) = 0 \iff \theta = \theta^*.$

Given that $p_M(\tilde{x})$ and $p_{\sigma 0}(\tilde{x})$ has the same support, it's clear that $L_{MSM} = 0$ would be equivalent to $L_{SM} = 0$. Due to the proof of the Theorem 2 in Hyvärinen (2005), we have $L_{SM}(\theta) \iff \theta = \theta^*$. Thus, $L_{MSM}(\theta) = 0 \iff \theta = \theta^*$.

$\square$

**Proposition 2.** $L_{MSM}(\theta) \smile L_{MDSM^*} = \mathbb{E}_{p_M(\tilde{x})q_{\sigma 0}(x|\tilde{x})} \parallel \nabla_{\tilde{x}} \log(q_{\sigma 0}(\tilde{x}|x)) + \nabla_{\tilde{x}} E(\tilde{x}; \theta) \parallel^2.$

We follow the same procedure as in Vincent (2011) to prove this result.

$$J_{MSM}(\theta) = \mathbb{E}_{p_M(\tilde{x})} \parallel \nabla_{\tilde{x}} \log(p_{\sigma_0}(\tilde{x})) + \nabla_{\tilde{x}} E(\tilde{x}; \theta) \parallel^2$$
$$= \mathbb{E}_{p_M(\tilde{x})} \parallel \nabla_{\tilde{x}} E(\tilde{x}; \theta) \parallel^2 + 2S(\theta) + C$$

$$S(\theta) = \mathbb{E}_{p_M(\tilde{x})} \langle \nabla_{\tilde{x}} \log(p_{\sigma_0}(\tilde{x})), \nabla_{\tilde{x}} E(\tilde{x}; \theta) \rangle$$
$$= \int_{\tilde{x}} p_M(\tilde{x}) \langle \nabla_{\tilde{x}} \log(p_{\sigma_0}(\tilde{x})), \nabla_{\tilde{x}} E(\tilde{x}; \theta) \rangle \, d\tilde{x}$$
$$= \int_{\tilde{x}} p_M(\tilde{x}) \langle \frac{\nabla_{\tilde{x}} p_{\sigma_0}(\tilde{x})}{p_{\sigma_0}(\tilde{x})}, \nabla_{\tilde{x}} E(\tilde{x}; \theta) \rangle \, d\tilde{x}$$
$$= \int_{\tilde{x}} \frac{p_M(\tilde{x})}{p_{\sigma_0}(\tilde{x})} \langle \nabla_{\tilde{x}} p_{\sigma_0}(\tilde{x}), \nabla_{\tilde{x}} E(\tilde{x}; \theta) \rangle \, d\tilde{x}$$
$$= \int_{\tilde{x}} \frac{p_M(\tilde{x})}{p_{\sigma_0}(\tilde{x})} \langle \nabla_{\tilde{x}} \int_x p(x) q_{\sigma_0}(\tilde{x}|x) dx, \nabla_{\tilde{x}} E(\tilde{x}; \theta) \rangle \, d\tilde{x}$$
$$= \int_{\tilde{x}} \frac{p_M(\tilde{x})}{p_{\sigma_0}(\tilde{x})} \langle \int_x p(x) \nabla_{\tilde{x}} q_{\sigma_0}(\tilde{x}|x) dx, \nabla_{\tilde{x}} E(\tilde{x}; \theta) \rangle \, d\tilde{x}$$
$$= \int_{\tilde{x}} \frac{p_M(\tilde{x})}{p_{\sigma_0}(\tilde{x})} \langle \int_x p(x) q_{\sigma_0}(\tilde{x}|x) \nabla_{\tilde{x}} \log q_{\sigma_0}(\tilde{x}|x) dx, \nabla_{\tilde{x}} E(\tilde{x}; \theta) \rangle \, d\tilde{x}$$
$$= \int_{\tilde{x}} \int_x \frac{p_M(\tilde{x})}{p_{\sigma_0}(\tilde{x})} p(x) q_{\sigma_0}(\tilde{x}|x) \langle \nabla_{\tilde{x}} \log q_{\sigma_0}(\tilde{x}|x), \nabla_{\tilde{x}} E(\tilde{x}; \theta) \rangle \, d\tilde{x} dx$$
$$= \int_{\tilde{x}} \int_x \frac{p_M(\tilde{x})}{p_{\sigma_0}(\tilde{x})} p_{\sigma_0}(\tilde{x}, x) \langle \nabla_{\tilde{x}} \log q_{\sigma_0}(\tilde{x}|x), \nabla_{\tilde{x}} E(\tilde{x}; \theta) \rangle \, d\tilde{x} dx$$
$$= \int_{\tilde{x}} \int_x p_M(\tilde{x}) q_{\sigma_0}(x|\tilde{x}) \langle \nabla_{\tilde{x}} \log q_{\sigma_0}(\tilde{x}|x), \nabla_{\tilde{x}} E(\tilde{x}; \theta) \rangle \, d\tilde{x} dx$$

Thus we have:

$$
\begin{aligned}
L_{MSM}(\theta) &= \mathbb{E}_{p_M(\tilde{\boldsymbol{x}})} \parallel \nabla_{\tilde{\boldsymbol{x}}} E(\tilde{\boldsymbol{x}}; \theta) \parallel^2 + 2S(\theta) + C \\
&= \mathbb{E}_{p_M(\tilde{\boldsymbol{x}})q_{\sigma_0}(\boldsymbol{x}|\tilde{\boldsymbol{x}})} \parallel \nabla_{\tilde{\boldsymbol{x}}} E(\tilde{\boldsymbol{x}}; \theta) \parallel^2 + 2\mathbb{E}_{p_M(\tilde{\boldsymbol{x}})q_{\sigma_0}(\boldsymbol{x}|\tilde{\boldsymbol{x}})} \langle \nabla_{\tilde{\boldsymbol{x}}} \log q_{\sigma_0}(\tilde{\boldsymbol{x}}|\boldsymbol{x}), \nabla_{\tilde{\boldsymbol{x}}} E(\tilde{\boldsymbol{x}}; \theta) \rangle + C \\
&= \mathbb{E}_{p_M(\tilde{\boldsymbol{x}})q_{\sigma 0}(\boldsymbol{x}|\tilde{\boldsymbol{x}})} \parallel \nabla_{\tilde{\boldsymbol{x}}} \log(q_{\sigma 0}(\tilde{\boldsymbol{x}}|\boldsymbol{x})) + \nabla_{\tilde{\boldsymbol{x}}} E(\tilde{\boldsymbol{x}}; \theta) \parallel^2 + C'
\end{aligned}
$$

So $L_{MSM}(\theta) \smile L_{MDSM^*}$.

$\square$

The above analysis applies to any noise distribution, not limited to Gaussian. but $L_{MDSM^*}$ has a reversed expectation form that is not easy to work with. To proceed further we study the case where $q_{\sigma_0}(\tilde{\boldsymbol{x}}|\boldsymbol{x})$ is Gaussian and choose $q_M(\tilde{\boldsymbol{x}}|\boldsymbol{x})$ as a Gaussian scale mixture (Wainwright and Simoncelli, 2000) and $p_M(\tilde{\boldsymbol{x}}) = \int q_M(\tilde{\boldsymbol{x}}|\boldsymbol{x})p(\boldsymbol{x})dx$. By Proposition 1 and Proposition 2, we have the following form to optimize:

$$
\begin{aligned}
L_{MDSM^*}(\theta) &= \int_{\tilde{\boldsymbol{x}}} \int_{\boldsymbol{x}} p_M(\tilde{\boldsymbol{x}})q_{\sigma_0}(\boldsymbol{x}|\tilde{\boldsymbol{x}}) \parallel \nabla_{\tilde{\boldsymbol{x}}} \log(q_{\sigma 0}(\tilde{\boldsymbol{x}}|\boldsymbol{x})) + \nabla_{\tilde{\boldsymbol{x}}} E(\tilde{\boldsymbol{x}}; \theta) \parallel^2 d\tilde{\boldsymbol{x}}d\boldsymbol{x} \\
&= \int_{\tilde{\boldsymbol{x}}} \int_{\boldsymbol{x}} \frac{q_{\sigma_0}(\boldsymbol{x}|\tilde{\boldsymbol{x}})}{q_M(\boldsymbol{x}|\tilde{\boldsymbol{x}})} p_M(\tilde{\boldsymbol{x}})q_M(\boldsymbol{x}|\tilde{\boldsymbol{x}}) \parallel \nabla_{\tilde{\boldsymbol{x}}} \log(q_{\sigma 0}(\tilde{\boldsymbol{x}}|\boldsymbol{x})) + \nabla_{\tilde{\boldsymbol{x}}} E(\tilde{\boldsymbol{x}}; \theta) \parallel^2 d\tilde{\boldsymbol{x}}d\boldsymbol{x} \\
&= \int_{\tilde{\boldsymbol{x}}} \int_{\boldsymbol{x}} \frac{q_{\sigma_0}(\boldsymbol{x}|\tilde{\boldsymbol{x}})}{q_M(\boldsymbol{x}|\tilde{\boldsymbol{x}})} p_M(\boldsymbol{x}, \tilde{\boldsymbol{x}}) \parallel \nabla_{\tilde{\boldsymbol{x}}} \log(q_{\sigma 0}(\tilde{\boldsymbol{x}}|\boldsymbol{x})) + \nabla_{\tilde{\boldsymbol{x}}} E(\tilde{\boldsymbol{x}}; \theta) \parallel^2 d\tilde{\boldsymbol{x}}d\boldsymbol{x} \\
&= \int_{\tilde{\boldsymbol{x}}} \int_{\boldsymbol{x}} \frac{q_{\sigma_0}(\boldsymbol{x}|\tilde{\boldsymbol{x}})}{q_M(\boldsymbol{x}|\tilde{\boldsymbol{x}})} q_M(\tilde{\boldsymbol{x}}|\boldsymbol{x})p(\boldsymbol{x}) \parallel \nabla_{\tilde{\boldsymbol{x}}} \log(q_{\sigma 0}(\tilde{\boldsymbol{x}}|\boldsymbol{x})) + \nabla_{\tilde{\boldsymbol{x}}} E(\tilde{\boldsymbol{x}}; \theta) \parallel^2 d\tilde{\boldsymbol{x}}d\boldsymbol{x} \quad (*) \\
&\approx L_{MDSM}(\theta)
\end{aligned}
$$

To minimize Equation (*), we can use the following importance sampling procedure (Russell and Norvig, 2016): we can sample from the empirical distribution $p(\boldsymbol{x})$, then sample the Gaussian scale mixture $q_M(\tilde{\boldsymbol{x}}|\boldsymbol{x})$ and finally weight the sample by $\frac{q_{\sigma_0}(\boldsymbol{x}|\tilde{\boldsymbol{x}})}{q_M(\boldsymbol{x}|\tilde{\boldsymbol{x}})}$. We expect the ratio to be close to 1 for the following reasons: Using Bayes rule, $q_{\sigma_0}(\boldsymbol{x}|\tilde{\boldsymbol{x}}) = \frac{p(\boldsymbol{x})q_{\sigma_0}(\tilde{\boldsymbol{x}}|\boldsymbol{x})}{p_{\sigma_0}(\tilde{\boldsymbol{x}})}$ we can see that $q_{\sigma_0}(\boldsymbol{x}|\tilde{\boldsymbol{x}})$ only has support on discret data points $\boldsymbol{x}$, same thing holds for $q_M(\boldsymbol{x}|\tilde{\boldsymbol{x}})$. because in $\tilde{\boldsymbol{x}}$ is generated by adding Gaussian noise to real data sample, both estimators should give results highly concentrated on the original sample point $\boldsymbol{x}$. Therefore, in practice, we ignore the weighting factor and use Equation 6. Improving upon this approximation is left for future work.

## B PROBLEM WITH SINGLE NOISE DENOISING SCORE MATCHING

To compare with previous method, we trained energy-based model with denoising score matching using one noise level on MNIST, initialized the sampling with Gaussian noise of the same level, and sampled with Langevin dynamics at $T = 1$ for 1000 steps and perform one denoise jump to recover the model's best estimate of the clean sample, see Figure B.1. We used the same 12-layer ResNet as other MNIST experiments. Models were trained for 100000 steps before sampling.

## C OVERFITTING TEST

We demonstrate that the model does not simply memorize training examples by comparing model samples with their nearest neighbors in the training set. We use Fashion MNIST for this demonstration because overfitting can occur there easier than on more complicated datasets, see Figure C.1.

## D DETAILS ON TRAINING AND SAMPLING

We used a custom designed ResNet architecture for all experiments. For MNIST and Fashion-MNIST we used a 12-layer ResNet with 64 filters on first layer, while for CelebA and CIFAR dataset

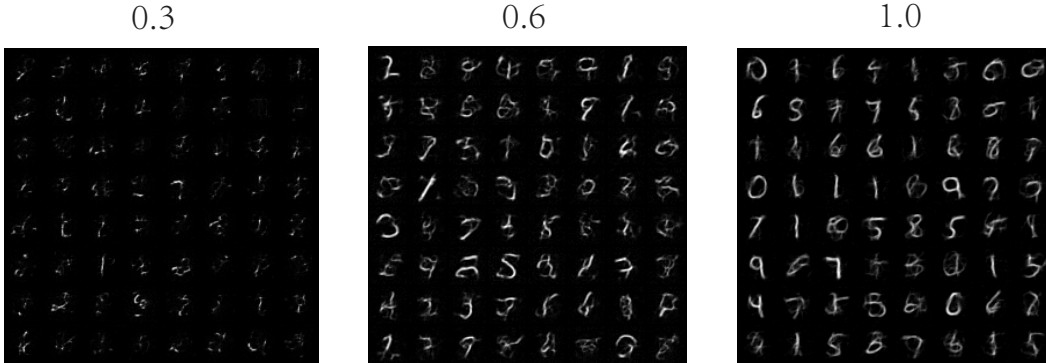

Figure B.1: Denoised samples from energy-based model trained with denoising score matching with single magnitude Gaussian noise on MNIST. Noise magnitude used in training is shown above samples.

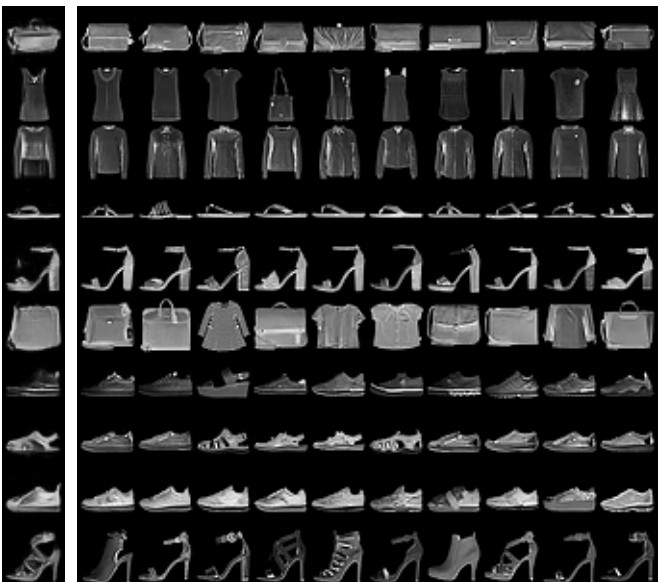

Figure C.1: Samples from energy-based model trained on Fashion MNIST (Left column) next to 10 (L2) nearest neighbors in the training set.

we used a 18-layer ResNet with 128 filters on the first layer. All network used the ELU activation function. We did not use any normalization in the ResBlocks and the filer number is doubled at each downsampling block. Details about the structure of our networks used can be found in our code release. All mentioned models can be trained on 2 GPUs within 2 days.

Since the gradient of our energy model scales linearly with the noise, we expected our energy function to scale quadratically with noise magnitude. Therefore, we modified the standard energy-based network output layer to take a flexible quadratic form (Fan et al., 2018):

$$E_{out} = (\sum_i a_i h_i + b_1)(\sum_i c_i h_i + b_2) + \sum_i d_i h_i^2 + b_3 \tag{12}$$

where $a_i, c_i, d_i$ and $b_1, b_2, b_3$ are learnable parameters, and $h_i$ is the (flattened) output of last residual block. We found this modification to significantly improve performance compared to using a simple linear last layer.

For CIFAR and CelebA results we trained for 300k weight updates, saving a checkpoint every 5000 updates. We then took 1000 samples from each saved networks and used the network with the lowest

FID score. For MNIST and fashion MNIST we simply trained for 100k updates and used the last checkpoint. During training we pad MNIST and Fashion MNIST to 32*32 for convenience and randomly flipped CelebA images. No other modification was performed. We only constrained the gradient of the energy function, the energy value itself could in principle be unbounded. However, we observed that they naturally stabilize so we did not explicitly regularize them. The annealing sampling schedule is optimized to improve sample quality for CIFAR-10 dataset, and consist of a total of 2700 steps. For other datasets the shape has less effect on sample quality, see Figure F.1 B for the shape of annealing schedule used.

For the Log likelihood estimation we initialized reverse chain on test images, then sample 10000 intermediate distribution using 10 steps HMC updates each. Temperature schedule is roughly exponential shaped and the reference distribution is an isotropic Gaussian. The variance of estimation was generally less than 10% on the log scale. Due to the high variance of results, and to avoid getting dominated by a single outlier, we report average of the log density instead of log of average density.

# E    EXTENDED SAMPLES AND INPAINTING RESULTS

We provide more inpainting examples and further demonstrate the mixing during sampling process in Figure E.1. We also provide more samples for readers to visually judge the quality of our sample generation in Figure E.2, E.3 and E.4. All samples are randomly selected.

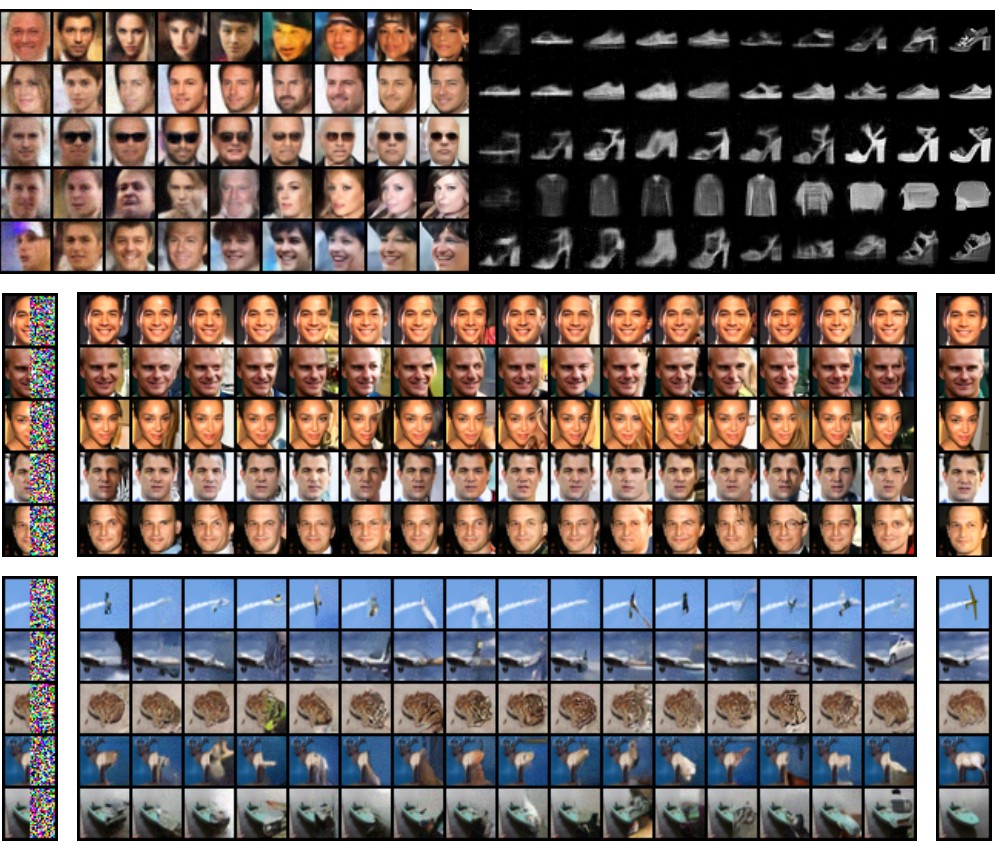

Figure E.1: Denoised Sampling process and inpainting results. Sampling process is from left to right.

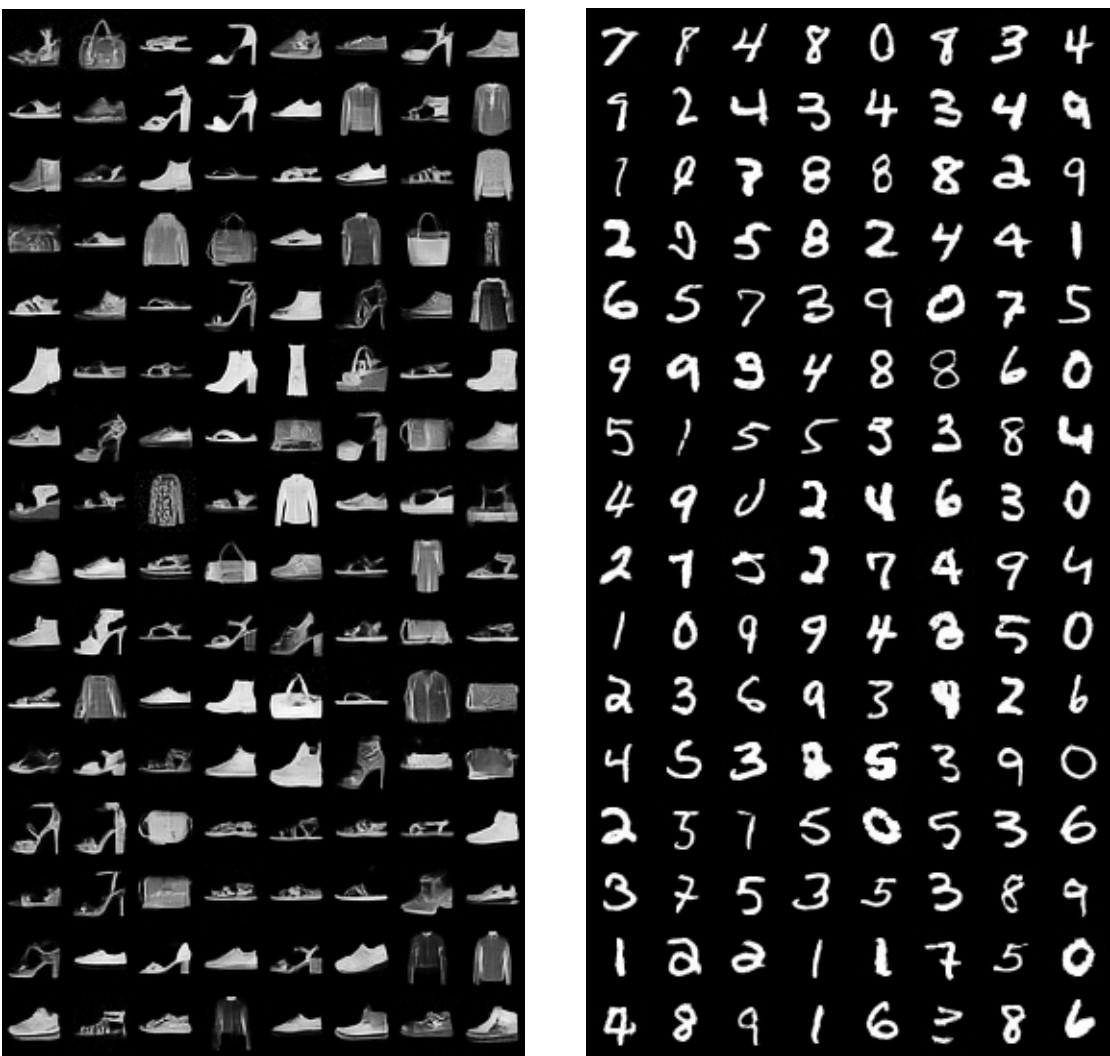

Figure E.2: Extended Fashion MNIST and MNIST samples

## F  SAMPLING PROCESS AND ENERGY VALUE COMPARISONS

Here we show how the average energy of samples behaves vs the sampling temperature. We also show an example of our model making out of distribution error that is common in most other likelihood based models (Nalisnick et al., 2019) Figure F.1.

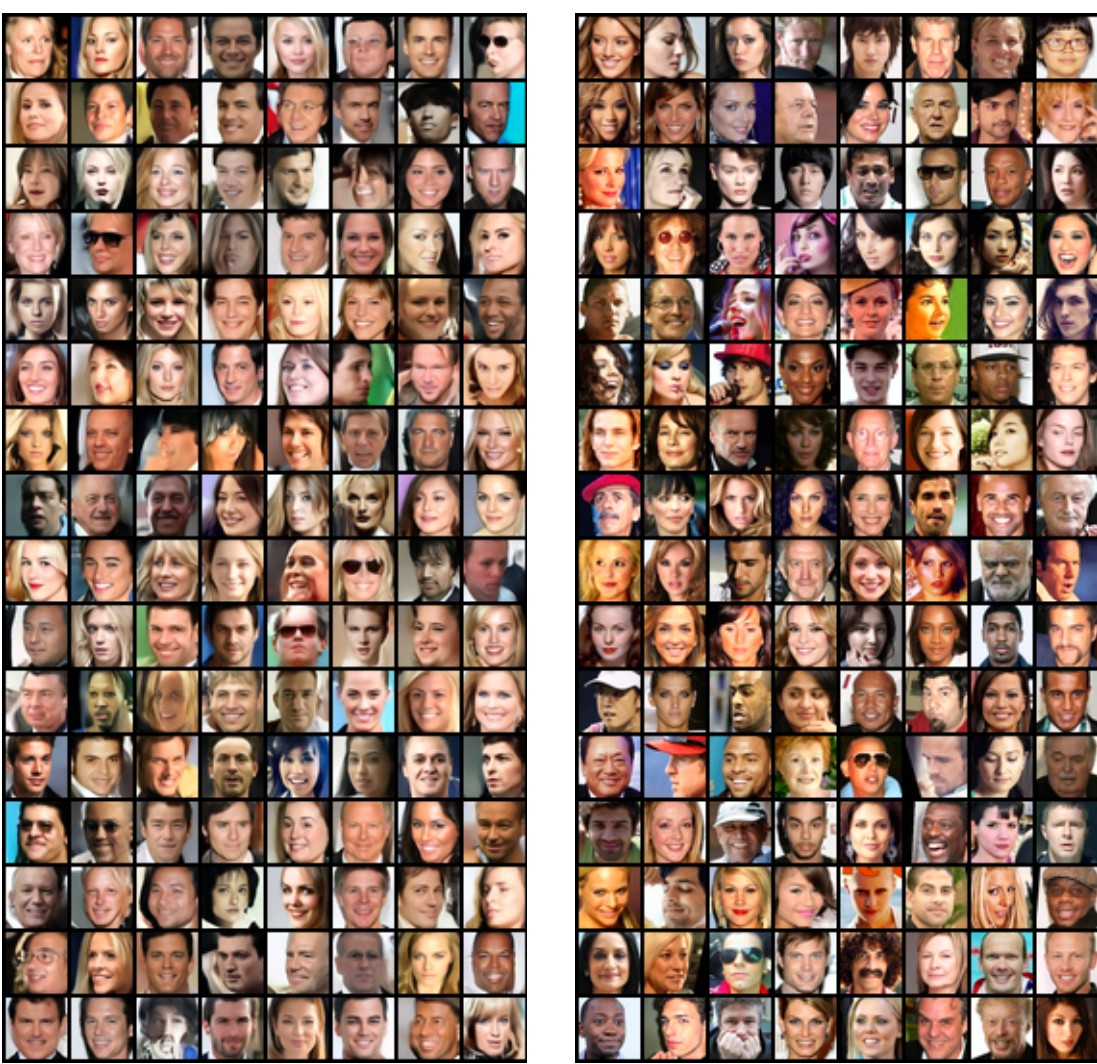

Figure E.3: Samples (left panel) from network trained on CelebA, and training examples from the dataset (right panel).

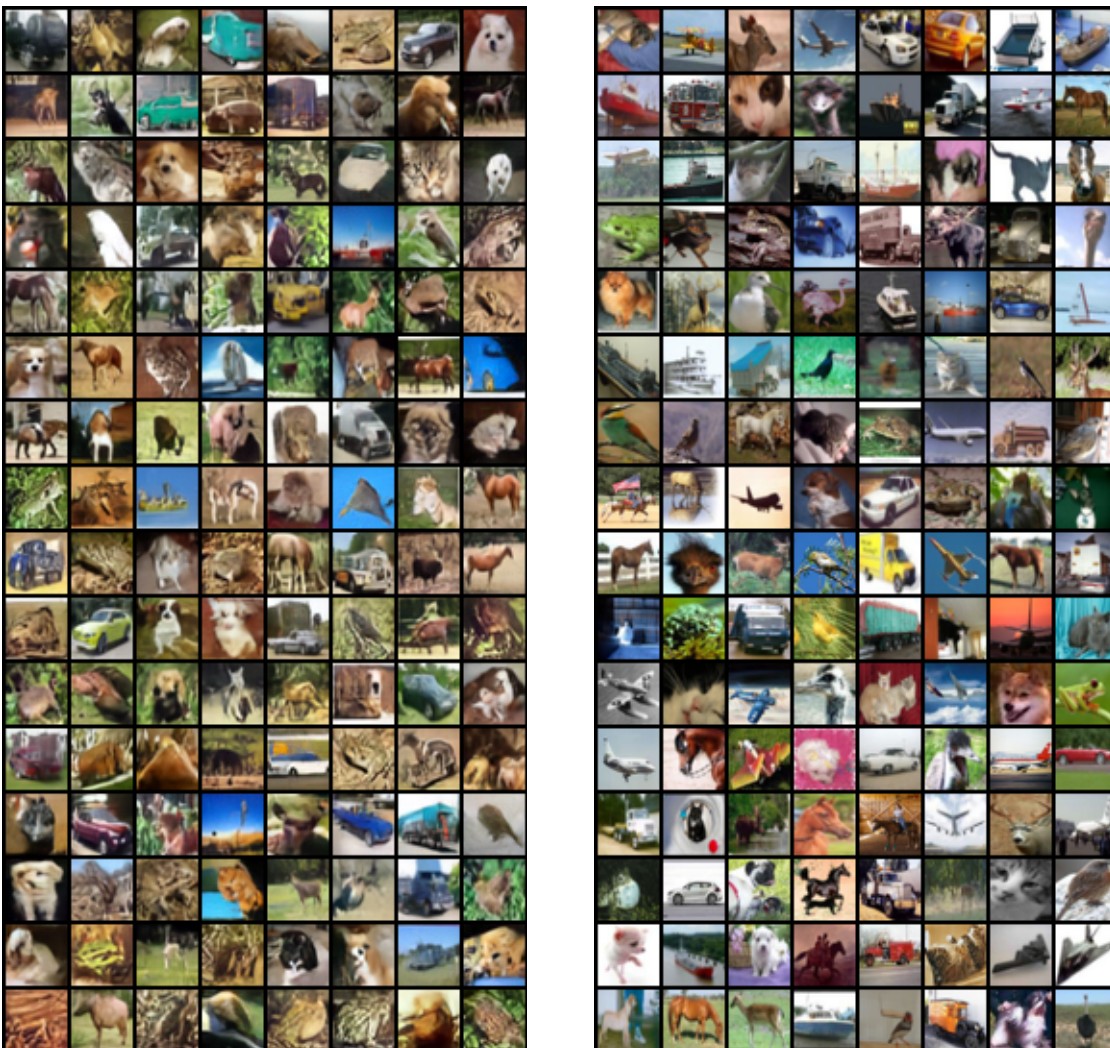

Figure E.4: Samples (left panel) from energy-based model trained on CIFAR-10 next to training examples (right panel).

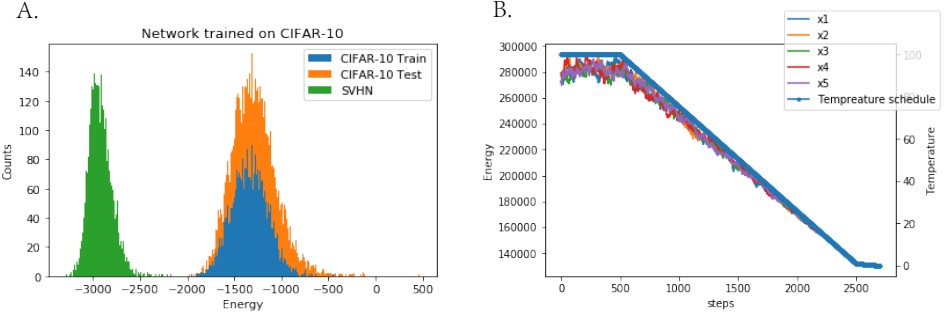

Figure F.1: A. Energy values for CIFAR-10 train, CIFAR-10 test and SVHN datasets for a network trained on CIFAR-10 images. Note that the network does not over fit to the training set, but just like most deep likelihood model, it assigns lower energy to SVHN images than its own training data. B. Annealing schedule and a typical energy trace for a sample during Annealed Langevin Sampling. The energy of the sample is proportional to the temperature, indicating sampling is close to a quasi-static process.

