# OpenReview forum: "Annealed Denoising score matching: learning Energy based model in high-dimensional spaces"
_ICLR.cc/2020/Conference — Reject_

### Official Review · AnonReviewer3 · 2019-10-21
**Official Blind Review #3**

**Rating:** 6

**Review:**

The paper proposes to learn an energy based generative model using an ‘annealed’ denoising score matching objective. The main contribution of the paper is to show that denoising score matching can be trained on a range of noise scales concurrently using a small modification to the loss. Compared to approximate likelihood learning of Energy based models the key benefit is to sidestep the need for sampling from the model distribution which has proven to be very challenging in practice. Using a slightly modified Langevin Sampler the paper further demonstrated encouraging sample qualities on CIFAR10 as measured by FID and IS scores.

Overall I think the paper is well motivated and written, experiments are sound with encouraging results that will be useful for further progress in training energy based models. I currently score the paper as a ‘weak accept’, the reason for not giving ‘accept’ is that I think the paper is closely related to Song & Ermin 2019 (see detailed comments below) - However i can be convinced to bump my score depending on the author feedback

Q1) I think you should elaborate more on how exactly your method is different from the NCSN model presented in Song & Ermon 2019? Especially.
Q1.1) Is your method similar to the NCSN except that you do linear scaling with temperature in the loss and train a joint model across all temperature scales?

Q1.2) In the related works section you claim that ‘[Song & Ermin] … this model learns p(xhat) for each T as a separate model’. Quickly reading through that paper i do not think that statement is accurate - I think they learn a model where the main difference is that it takes T as input instead of scaling the gradient term in the loss?

Q1.3 )Do you have any intuition for why they seem to get slightly better results than the one you obtain in your paper? Is it simply architecture/training details that differ or something more ‘fundamental’?


Q2) In relation to the Score matching objective.
Q2.1) In eq (4) it is not completely clear to me what the motivation for linear scaling in T is. Can you elaborate on what you mean with ‘We borrow intuition from physics and simply set E_T(xhat) = E(xhat)/T ...’?
In relation to the above Can you clarify which part of your results holds for Gaussian noise and which holds in general.

Q2.2) For the gaussian case I think linear scaling as done in eq(5) is sensible, however for arbitrary noise distributions linear scaling is akin to a first order approximation (which might be inaccurate across a range of different noise levels)?
Minor: I think it would ease the reading of the paper if you showed the derivation (in appendix) that Eq (1) and Eq(2) are equivalent.

Minor Comment: Learning generative models using denoising have also been explored in [Soenderby 2016]. Here the difficulties of different noise scales was also found and explored but (importantly) not solved.

[Song & Ermon]. Generative Modeling by Estimating Gradients of the Data Distribution
[Soenderby 2016]: Amortised map inference for image super-resolution


**Experience Assessment:**

I have published one or two papers in this area.

**Review Assessment: Checking Correctness Of Derivations And Theory:**

I assessed the sensibility of the derivations and theory.

**Review Assessment: Checking Correctness Of Experiments:**

I carefully checked the experiments.

**Review Assessment: Thoroughness In Paper Reading:**

I read the paper thoroughly.

---

> ### Author Response · Authors · 2019-11-10
> **Response to reviewer #3**
>
> Thank you for your review, comments and encouraging feedbacks!
>
> We have revised the presentation of our algorithm accordingly to better reflect the essence of our algorithm and the conceptual difference between our method and that of Song&Ermon 2019.
>
> For answer to Q1 1) , Q1 3) and part of Q2 1) please refer to the general response and section 3 of the revision of the paper.
>
> Regarding Q1 2). Indeed NCSN model take noise scale as input but is not a set of completely separate models, so our statement is not entirely accurate. We have thus updated the relevant statements in our paper. The most essential difference between our model and NCSN is that NCSN learns score of a series of different distributions while our method learns only one distribution.
>
> Regarding Q2 1) and 2) The original motivation for our model causes unnecessary confusions, therefore, we have revised our presentation in the updated manuscript. In the revision, we have clarified which part of our algorithm applies generally and which part applies only to Gaussian noise. Essentially equation 4) and 5) apply to any noise distribution, but to make the approximation between equation 5) and 6), one has to choose specific distribution to take the score average over, which will require specific knowledge about the noise distribution.
>
> Thanks
>
> References
> Y Song, S Ermon. Generative Modeling by Estimating Gradients of the Data Distribution. NeurIPS 2019.

---

### Official Review · AnonReviewer1 · 2019-10-23
**Official Blind Review #1**

**Rating:** 3

**Review:**

########Updated Review ###########

I would like to thank the author(s) for their reply, which I have carefully read and it partly addresses my original concerns. Still, as agreed by all three reviewers, this paper might not be a significant step up compared with [1]. I am raising my point to weak reject to reflect my updated belief. I think this paper needs a bit more highlights to pass the threshold.

###############################


This paper tries to address the problem of non-parametric maximal likelihood estimation via matching the score function wrt data. It is a clear rejection due to its significant overlap with the recent NeurIPS publication [1]. The author(s) have failed to clarify how their proposal differs from [1] in a significant way. From what I can tell after a quick read, both papers tried to training the score function using the denoising auto-encoder, amortized through a neural network, strategically annealed with a sequence of different noise levels, sampled with the Langevin scheme. I put two papers side-by-side and you can visually tell the uncanny resemblance.  Additionally, the proposed model does not outperform that from [1] (see Table 1). I am also not happy about the misleading statement in the abstract that this work "assign likelihood to test data", which is actually performed by AIS.  Section 2.2 is particularly problematic. The assumption of "data approximately uniformly distributed on the manifold" is outrageous, which basically invalidates the need for density estimation because of the uniformity. The 1/f power law characteristic is irrelevant to the likelihood estimation problem, and the statements are both heuristic & misleading.

[1] Y Song, S Ermon. Generative Modeling by Estimating Gradients of the Data Distribution. NeurIPS 2019.

**Experience Assessment:**

I have published one or two papers in this area.

**Review Assessment: Checking Correctness Of Derivations And Theory:**

I assessed the sensibility of the derivations and theory.

**Review Assessment: Checking Correctness Of Experiments:**

I assessed the sensibility of the experiments.

**Review Assessment: Thoroughness In Paper Reading:**

I made a quick assessment of this paper.

---

> ### Author Response · Authors · 2019-11-10
> **Response to reviewer #1**
>
> Thank you for your response and please kindly allow us to explain ourselves better.
>
> Your major concern, the overlap between our paper and Song & Ermon 2019, and the slight underperformance of our model,  has been addressed in the general response. Please also refer to section 3 of our updated manuscript for a better presentation of our proposed model.
>
> Regarding your concern about the statement “assign likelihood to data”. In our opinion, energy-based models should be considered likelihood-based as energy value represents unnormalized log likelihood. After partition function has been estimated by methods such as AIS and reverse AIS, normalized log-likelihood can be obtained for any data point.
>
> We have also revised section 2 so that it no longer contains speculative claims.
>
> Thanks
>
> References
> Y Song, S Ermon. Generative Modeling by Estimating Gradients of the Data Distribution. NeurIPS 2019.

---

### Official Review · AnonReviewer2 · 2019-10-23
**Official Blind Review #2**

**Rating:** 3

**Review:**

This paper presents a method of learning of energy based models using denoising score matching. This technique has been used before but only with limited success. The authors hypothesize that this is due to the fact that the matching was only performed over a single noise scale. The main idea of this work is to employ a range of scales to learn a single energy function. This trick helps to alleviate the problem of noisy samples concentrating in a low-volume region of the ambient space.

It seems that the paper draws significant inspiration from the work by Song & Ermon, 19. The difference between the two appears to be minor:
1) The density is represented as a Boltzman distribution and therefore the score function is reduced to the gradient of the energy function (this has been done before)
2) Instead of conditioning the energy on the noise level the authors propose to use explicit scaling by the inverse temperature

Pros:
+ The paper is mostly well-written.
+ I think Section 2 does a good job at illustrating challenges in training energy-based models using denoising score matching with a single noise scale.
+ As compared to (Song & Ermon, 19) using the Boltzman distribution ensures that the learned score is an actual conservative vector field. Arguably, learning an image to scalar network is easier than learning an image to image one.
+ Samples from the model are of competitive visual quality.

Cons:
- Scaling energy by the inverse temperature seems to be one of the most important aspects of the paper but is only justified by “intuition from physics”. I’m not entirely sure that this is a valid assumption. In contrast, (Song & Ermon, 19) don’t put any hard constraints on the values of the score for different noise levels besides that they are produced by a single conditional network. I would appreciate if the authors discussed that difference in more detail.
- The authors don’t provide any analysis as to whether the annealed Langevin MC procedure leads to the samples from the right distribution.
- The quantitative results don’t seem to be better (actually, they are worse) than those from (Song & Ermon, 19).

Notes/questions:
* Abstract: “unmormalized” -> “unnormalized”
* Section 2.1, (1): \tilde{x} -> \tilde{\mathbf{x}}
* Section 2.2, paragraph 2: What does superscript C mean in the noisy manifold? Never defined.
* Section 2.2, paragraph 4: “some example” -> “some examples” (?)
* Section 3, paragraph 1: “CIFAT-10” -> “CIFAR-10”
* Section 4, paragraph 2: “for each T as a separate model”. I don’t think this is a correct statement. (Song & Ermon, 19) use a single conditional model for all the noise levels.
* Section 4, paragraph 2: “does not rely on explicit receive noise magnitude” -> “does not rely on receiving noise magnitude explicitly” (?) I also don’t quite understand this entire sentence. Does the model really infer the noise magnitude from a given image? It seems like in Equation (7) there is an assumption that the temperature T is equal to 1. I don’t feel like there is a lot of difference between the proposed model and (Song & Ermon, 19) when it comes to supplying noise information. I’d appreciate if the authors could clarify that bit for me.

My main concern about this paper is that it doesn’t seem like a big step from its starting point (Song & Ermon, 19). The modifications are shown to work empirically but don’t result in a significantly better model. Moreover, I feel like the paper could do a better job at justifying those changes. I’m giving a borderline score but willing to increase it if the authors address my questions.

**Experience Assessment:**

I do not know much about this area.

**Review Assessment: Checking Correctness Of Derivations And Theory:**

I assessed the sensibility of the derivations and theory.

**Review Assessment: Checking Correctness Of Experiments:**

I assessed the sensibility of the experiments.

**Review Assessment: Thoroughness In Paper Reading:**

I read the paper at least twice and used my best judgement in assessing the paper.

---

> ### Author Response · Authors · 2019-11-10
> **Response to reviewer #2**
>
> Thank you for your review, comments and suggestions for corrections!
>
> We realized that the original presentation of our algorithm is misleading and have revised our paper accordingly to better present the core idea of our model. Please kindly take a look at section 3 of the updated manuscript.
>
> We addressed the relationship between our work and that of Song&Ermon 2019 in the general response, which should also clarify your last question: “ Does the model really infer the noise magnitude from a given image?”. We also discussed possible reasons for the slight underperformance of our model.
>
> Regarding the concern about the convergence of annealed Langevin dynamics. We would like to note that it is a well-known classical result that under Langevin dynamics, the probability density of samples evolves according to the Fokker-Plank equation, which then have Boltzmann distribution p(x) = exp(-E(x)/T)/Z as equilibrium solution. This applies to any constant temperature T. However, according to Neal 2001, annealing process is a heuristic method and there is no theoretical guarantee that an annealing process will produce fair samples from the final distribution, although importance sampling technique can be used if an unbiased average of some function is needed (Neal 2001).
>
> References
> Y Song, S Ermon. Generative Modeling by Estimating Gradients of the Data Distribution. NeurIPS 2019.
>
> RM Neal. Annealed Importance Sampling. Statistics and computing, 2001.

---

### Author Response · Authors · 2019-11-10
**Response to all reviewers**


We thank all the reviewers for their efforts on evaluations and helpful comments!

First, we would like to address a major concern shared by all three reviewers: a potential overlap between our work and Song&Ermon 2019. We acknowledge that the original paper could evoke this impression. We first explain the difference between Song and Ermons’ and our model and then describe the changes in the manuscript to address this issue.
1) Model differences:
The NCSN model is trained with multiple noise levels and learns a score function conditioned on noise level. In other words, the deep network in the NCSN model learns to map a tuple of a data point and noise level to a score vector.

The most important difference of our model to the NCSN model is that it is an energy-based model. In other words, the deep learning network in our model maps a data point to an energy value. Thus, the mapping uses the noise level in the data point implicitly, rather than receiving the noise level as an additional input parameter.

This difference is reflected in the corresponding objective functions. Both objectives consist of a weighted sum of expectation values of an L2 distance and look superficially similar. But note that in the NCSN objective each L2 term in the sum contains \sigma_i , a parameter that changes with i.  In contrast, each L2 term in our model contains the same sigma_0, the parameter of the fixed Parzen window. Further note, that the neural network in NCSN model has \sigma_i as argument, in addition to the data point, whereas the sole argument in the neural network of our model is the data point.

As a result of this difference, our model can perform single step denoising over all noise scales without prior information about the noise magnitude. Further, our model is directly a density function of the data whereas it is not straight-forward how to convert the noise conditioned score of the NCSN model into a density.


2) We have thoroughly rewritten the abstract and body of the paper to make our contribution easily accessible.

It is now clearly acknowledged that the NCSN model is the first generative model based on denoising score matching that uses noise with multiple levels in the training to provide state-of-the-art performance in sample synthesis of high dimensional datasets.

The two contributions of our work are now clearly described.

1) An energy based model providing state-of-the-art performance (among energy-based models) in sample synthesis of high dimensional data.

2) Starting from the manifold hypothesis also used by Song & Ermon, we provide theoretical argument along with empirical evidence on why training with multiple noise levels is required for modeling high-dimensional data.
This contribution has been recognized by Reviewer 2.


Second, all reviewers asked why our model performs slightly inferior on sample generation than the NCSN.

Other than the difference in model architecture and fine tuning, one plausible reason for the slight underperformance of our model is the following: Our model is more parsimonious, but during derivation of the objective function (see section 3.1 of the new version of the paper), an approximation is needed which may reduce performance.

Additionally, the NCSN’s output is a vector that, at least during optimization, does not always have to be the derivative of a scalar function. For a vector field of dimension n, being the gradient of a scalar function amounts to satisfying n*(n-1)/2 partial differential equations as constraints, as the high-dimensional equivalence of the curl must be 0. For the CIFAR-10 dataset this amounts to more than 1500 constraints per pixel. In contrast, in our model the network output is a scalar function. Thus it is possible that the NCSN model performs better because it explores a larger set of functions during optimization.


References
Y Song, S Ermon. Generative Modeling by Estimating Gradients of the Data Distribution. NeurIPS 2019.

---

### Decision · Program_Chairs · 2019-12-19

**Decision:**

Reject

**Comment:**

This paper presents a variant of the Noise Conditional Score Network (NCSN) which does score matching using a single Gaussian scale mixture noise model. Unlike the NCSN, it learns a single energy-based model, and therefore can be compared directly to other models in terms of compression. I've read the paper, and the methods, exposition, and experiments all seem solid. Numerically, the score is slightly below the cutoff; reviewers generally think the paper is well-executed, but lacking in novelty and quality of results relative to Song & Ermon (2019).